# Exploring Effects of Graphene and Carbon Nanotubes on Rheology and Flow Instability for Designing Printable Polymer Nanocomposites

**DOI:** 10.3390/nano13050835

**Published:** 2023-02-23

**Authors:** Rumiana Kotsilkova, Sonia Tabakova

**Affiliations:** 1Open Laboratory on Experimental Micro and Nano Mechanics, Institute of Mechanics, Bulgarian Academy of Sciences, Acad. G. Bontchev, Bl.4, 1113 Sofia, Bulgaria; 2Institute of Mechanics, Bulgarian Academy of Sciences, Acad. G. Bontchev, Bl.4, 1113 Sofia, Bulgaria

**Keywords:** PLA nanocomposite, graphene nanoplatelets, carbon nanotubes, shear banding flow, Herschel-Bulkley model, engineering design, 3D printing

## Abstract

Nowadays, a strong demand exists for printable materials with multifunctionality and proper rheological properties to overcome the limitations to deposit layer-by-layer in additive extrusion. The present study discusses rheological properties related to the microstructure of hybrid poly (lactic) acid (PLA) nanocomposites filled with graphene nanoplatelets (GNP) and multiwall carbon nanotubes (MWCNT) to produce multifunctional filament for 3D printing. The alignment and slip effects of 2D-nanoplatelets in the shear-thinning flow are compared with the strong reinforcement effects of entangled 1D-nanotubes, which govern the printability of nanocomposites at high filler contents. The mechanism of reinforcement is related to the network connectivity of nanofillers and interfacial interactions. The measured shear stress by a plate–plate rheometer of PLA, 1.5% and 9% GNP/PLA and MWCNT/PLA shows an instability for high shear rates, which is expressed as shear banding. A rheological complex model consisting of the Herschel–Bulkley model and banding stress is proposed for all considered materials. On this basis, the flow in the nozzle tube of a 3D printer is studied by a simple analytical model. The flow region is separated into three different regions in the tube, which match their boundaries. The present model gives an insight into the flow structure and better explains the reasons for printing enhancement. Experimental and modeling parameters are explored in designing printable hybrid polymer nanocomposites with added functionality.

## 1. Introduction

Materials extrusion (ME), including Fused Deposition Modeling and Fused Filament Fabrication, according to ISO/ASTM 52900:2015, is among the most widely employed Additive Manufacturing technologies for the production of parts with complex geometries for specific uses. The ME belongs to a CAD-controlled melt extrusion through a nozzle to build objects by layer-to-layer deposition, named 3D printing [1,2]. Proper rheological properties are needed to overcome the limitations related to deposit the material, therefore only relatively few polymers and composites can be easily processed with this technology. The control of materials’ rheology may provide a flexible manufacturing route to fabricating 3D-printed parts with a good resolution [2,3]. The importance of rheology and shear thinning behavior for identifying the 3D printability of polymer nanocomposites via ME is widely discussed [4,5]. The main finding is that the melt extrusion process needs careful rheological measurements, modeling, simulation and optimization to select the best processing conditions and the best material properties towards successful 3D printing, as well as to allow process and instrument development [2]. However, the relationship between the rheological characteristics of materials and 3D printability needs further investigation to establish a connection between the rheology, printing parameters and the flow profile [5,6].

Hybrid polymer nanocomposites incorporating 1D and 2D carbonaceous nanofillers are intensively studied for the development of multifunctional materials for 3D printing application with robust electrical, electromagnetic, thermal and mechanical properties [7,8,9,10]. The enhancing of thermal conductivity and diffusivity, electromagnetic shielding efficiency, electrical conductivity and mechanical reinforcement of nanocomposites by graphene-based additives open frontiers for a variety of applications in high-power electronics, sensors, actuators, photovoltaics, energy storage, low-frequency energy converters, etc. [11,12,13,14,15,16]. The physical mechanism of nanocomposite properties improvement by the addition of graphene and carbon nanotubes is usually related to the enormous surface area of nanofillers, the interfacial filler-polymer interactions, and the percolation network of fillers in the polymer, ensuring an efficient transmission of load and functionalities between the filler and the polymer [7,8,9,10]. Due to the complexity of composition and processing, however, rheological properties affected by the filler types, size and shape, degree of dispersion, polymer-filler and filler-to-filler interactions are essential in order to achieve control of the 3D-printing process and to tailor the structure–properties relations of the printed parts [5,7,17]. The flow of polymers and nanocomposites in additive extrusion is not yet fully understood, as it involves many complex phenomena, such as phase transition, shear and time-temperature dependent viscoelastic behavior, and other rheological complexity, which may result in flow instabilities during the ME process [18,19].

Many complex fluids undergo flow instabilities, resulting in heterogeneous ‘‘shear banded’’ states which display oscillations or irregular fluctuations [20,21]. Shear banding, either steady or transient, is now well-established in a variety of soft materials, including worm-like micelles, liquid crystals, triblock copolymers, star polymers, entangled polymer solutions and melts, yield stress fluids, colloidal suspensions, etc. [22,23,24,25]. Particularly, in polymeric materials, shear banding is a phenomenon whereby regions of different shear rates, called “bands,” can coexist in a system with sharp interfaces between them; for example, liquid-like bands flow to a solid-like region or a highly aligned phase adjacent to a less aligned one [23]. Banding is often initiated by instability, and it results in heterogeneous flows, where two different shear rates coexist for a given shear stress, in which shear bands are organized along the flow gradient direction [24]. From a microstructure aspect, the shear banding is usually driven by the lower viscosity, when the polymer chains are aligned more parallel to each other in the flow, which pushes the stress slope to negative at high shear rates [25,26]. Thus, the entangled micellar system is regarded as extreme shear thinning, where the positive-to-negative transition of the stress slope by increasing the concentration distinguishes shear thinning from shear banding behavior at high shear rates [25]. There is also a significant amount of theoretical work on modeling shear banding polymeric fluids in different flow geometries [20,21,22,23,24,25,26,27].

A limited number of publications discussed the flow instabilities during 3D printing (ME), where the filament is advanced to the hot end, which consists of a melting zone and a nozzle that extrudes the material [2,19,28]. It is generally accepted that two factors are important here: elongation flow properties under the pressure drop and flow instabilities that may develop [2]. Two extrusion regimes are identified in the hot-end flow: a linear regime with a stable flow, and a non-linear regime, where the force oscillates and increases rapidly as a function of the feeding rate, leading to unstable extrusion [19]. Zhu et al. [28] observed the shear banding of well-entangled polymer melts at the die entrance during material extrusion under a controlled rate or controlled pressure. The origin of the shear banding in linear chain melts is proposed as the localized chain disentanglement in the entangled polymer melt under pressure in extrusion.

Few studies reported shear banding in the shear flow of aqueous suspensions of graphene oxide (GO) platelets at filler content above the percolation threshold, originated from the coexistence of solid-like and fluid regions, below critical shear rate [29,30]. As a physical interpretation of this effect, the alignment of the platelets along the direction of flow above the yield stress is assumed, which significantly reduces the viscosity and causes shear-thinning behavior to arise. In our previous study [5] on the 3D printability of polymer nanocomposites with carbonaceous fillers, graphene and carbon nanotubes, we proposed a flow instability in the printing nozzle due to the “elastic turbulence” caused by a flow with a high Weissenberg number and very low Reynolds number for the highly filled systems with carbon nanotubes. However, publications on flow instabilities associated with shear banding during the 3D printing (ME) of polymer nanocomposites and how this phenomenon is affected by anisotropic nanofillers are not found in the scientific literature.

In the present study, we focus on the extreme shear thinning polylactic acid (PLA) nanocomposites filled with multiwall carbon nanotubes (MWCNT) and graphene nanoplatelets (GNP), at high shear rates of flow, in the material extrusion additive manufacturing process. We suspect that the negative slope of the flow curve in the shear-thinning region is attributed to instability, expressed as banding. An analytical model for shear stress with banding is proposed for the flow in the 3D-printing nozzle. The experimental and analytical flow parameters are proposed for engineering design of hybrid polymer nanocomposites towards good printability and multifunctionality.

## 2. Materials and Methods

The Ingeo™ Biopolymer PLA-3D850 from Nature Works (Plymouth, MN, USA), with MFR 7–9 g/10 min (210 °C, 2.16 kg); graphene nanoplatelets (TNIGNP), from Times Nano, China and multiwalled carbon nanotubes (MWCNT, NC7000), from Nanocyl, Belgium are used in this study. The basic characteristics of nanofillers are summarized in Table 1.

Nanocomposites of 1.5 wt.% and 9 wt.% GNPs and MWCNTs in PLA polymer are prepared for this study, as the extreme cases of the filler content. The 1.5 wt.% GNP and/or MWCNT is associated with the start of the enhancement in mechanical reinforcement, physical properties and functionality of nanocomposites. This filler content is around the electrical percolation threshold estimated in our previous studies [7,10], such as EPT ≥ 1.5 wt.% for MWCNT/PLA and EPT ≥ 3 wt% for GNP/PLA nanocomposites. Moreover, the 9 wt% filler content was found to be the limit of printability for the MWCNT/PLA nanocomposites. Nanocomposites are fabricated by a twin screw extruder, where the 9 wt.% masterbatches of GNP/PLA and MWCNT/PLA are produced with a screw speed of 40 rpm, at temperatures 170–180 °C. Then, the formulation with 1.5 wt.% fillers is prepared by dilution of the masterbatch with the neat PLA through a second extrusion run. The filament for 3D printing is fabricated from the nanocomposite pellets by a single screw extruder at 10 rpm within the same temperature range, followed by quenching in a water bath at 60 °C. The filament diameter of 1.75 ± 0.5 mm is controlled by a laser counter during the extrusion process.

The printability of the filaments is tested with a German RepRap X-400 Pro 3D printer (FDM), having a printing speed limit of *V* = 10–150 mm/s. Complex-shaped samples are printed by layer-to-layer deposition at the following printing conditions: nozzle temperature 220 °C, table temperature 65 °C, printing speed *V* = 10 mm/s, nozzle diameter *D* = 0.5 mm, layer thickness *L* = 0.3 mm and printing density 100% infill.

Bright field TEM analysis is performed on a FEI TECNAI G12 Spirit-Twin (LaB6 source, Hillsboro, OR, USA), operating with an acceleration voltage of 120 kV. Thin slides of 80 nm thickness are cut from the sample and placed on a holder for further observation. SEM visualization is made by using a field emission scanning electron microscope (FESEM, mod. LEO 1525, Carl Zeiss SMT AG, Oberkochen, Germany). 3D-printed samples are immersed in liquid nitrogen, then cut at the cross-section and finally coated with gold (layer thickness 250 A) using a sputter coater (mod. B7341, Agar Scientific, Stansted, UK). The surface and the cross-section of the sample are visualized at different magnifications.

Rheological properties of nanocomposites are measured by a rotational rheometer AR-G2, TA Instruments, USA, using electrical heating parallel plate geometry, with a 25 mm diameter and measurement gap of 1 mm. The steady-state flow test in the shear rate range of 0.05–100 s^−1^ is performed at a temperature of 220 °C.

## 3. Results

### 3.1. Microstructure Observation

Scanning electron microscopy (SEM) analysis was performed for the 3D-printed PLA-based nanocomposite samples at 9 wt.% filler content. Figure 1a visualizes the surfaces of the printed GNP/PLA top layer, which is constructed of parallel strips with some voids between them, obtained at these printing conditions. The SEM micrographs in Figure 1b,c present the cross section of the composite samples of graphene nanoplatelets and carbon nanotubes to get information regarding the dispersion and interactions between the matrix and the fillers. The fracture surfaces of the GNP/PLA composite (Figure 1b) show an anisotropic type structure formed by interacted and aligned graphene nanoplatelets. In contrast, an isotropic dense structure is visible over the entire surface of the MWCNT/PLA cross-section (Figure 1c), built as a homogeneous network of entangled nanotubes.

Figure 2a–d displays the TEM micrographs for the composites GNP/PLA and MWCNT/PLA with 1.5 and 9 wt.% filler content. The 1.5% CNP/PLA micrograph (a) shows partly exfoliated graphene particles, not contacting each other, which is a typical structure before the percolation threshold. However, the microstructure of 1.5 wt.% MWCNT/PLA (b) consists of well-dispersed particles. A totally different dispersion microstructure is obtained for 9 wt.% filler content (Figure 2b,c), depending on the composition. The micrograph of 9 wt.% GNP/PLA (c) shows large graphene aggregates of size above 500 nm to a few microns, which have an anisotropic shape and construct a percolation network. In contrast, Figure 2d demonstrates a dense percolation structure of entangled nanotubes in the 9 wt.% MWCNT/PLA composite, single nanotubes and small aggregates which are contacting to form a percolation network. The presence of percolated network above the percolation threshold of EPT ≥ 1.5 wt.% and EPT ≥ 3 wt% for the MWCNT and the GNP, respectively, in PLA nanocomposites was estimated in our previous studies [5,7,10] using electrical and rheological measutements with varying filler content in a wide range.

The micrographs in Figure 2a–d present a typical structure of the hybrid polymer nanocomposites, having well dispersed nanofiller and good interfacial filler–polymer interactions. The hybrid nanocomposite structure depends strongly on the type, shape, dispersion and concentration of the filler, which can significantly affect the flow behaviour during additive processing, like 3D printing (ME).

### 3.2. Rheological Behavior of Hybrid Polymer Nanocomposires

Yield stress fluids can be considered an extreme example of shear thinning [26]. At low shear rates, the viscosity is very high, and the material behaves as a solid. Above a critical shear rate (or range of rates), an effectively solid-to-liquid reversible transition appears, which may have noteworthy applications in 3D printing [26,31,32]. The rheological behavior of hybrid polymer nanocomposite melts based on PLA filled with MWCNTs and GNPs, approaches the limits of a yield stress fluid.

In Figure 3a,b the flow curve, viscosity vs. shear rate is plotted for the two types of nanocomposites: (a) MWCNT/PLA and (b) GNP/PLA, at filler contents of 0, 1.5 and 9 wt.%, in the shear rate range 0.05–100 s^−^^1^, at 220 °C. Considering the MWCNT/PLA nanocomposites in Figure 3a, at low shear rates the neat PLA shows a Newtonian plateau of the zero-shear viscosity. While the two MWCNT/PLA systems at 1.5 and 9 wt% nanotube contents behave as typical yield stress fluids with very high viscosity at low shear rates, having values of 10^4^ and 10^6^ Pa.s, respectively, compared to 10^2^ Pa.s at ($\dot{\gamma }$) = 0.05 s^−1^, for the neat PLA. This effect is associated with the percolation network structure of entangled nanotubes with varying density, depending on the filler contents, as shown in Figure 2b,d. Such a dense structure of nanotubes, particularly for 9 wt.% MWCNT, can suppress the flow in the 3D printing nozzle. At high shear rates of ($\dot{\gamma }$) ≥ 20 s^−1^, a shear thinning appears for both the neat PLA and the MWCNT/PLA nanocomposites. Generally, the viscosity of the nanocomposites in the shear thinning range is higher than that of the neat PLA, and increases by increasing the filler content. In the shear thinning range, the viscosity slope of 1.5 wt.% MWCNT system is similar to that of the neat PLA, while the slope for 9 wt.% MWCNT system is slightly higher; but importantly, all slopes are negative. This effect has a microstructure origin, related to the disentanglement of both the entangled polymer chain and the MWCNTs in the flow field at high shear rates.

In contrast, the GNP/PLA nanocomposites in Figure 3b demonstrate quite different flow behavior compared to MWCNT composites at the same filler contents, due to the different size and shape effects of the two nanofillers. Thus, at low shear rates, the 1.5 wt.% GNP/PLA system demonstrates a Newtonian plateau, similar to the neat PLA, but with higher viscosity due to the presence of nanofiller. While at 9 wt.% GNP content the system behaves as a yield stress fluid, with a decade higher viscosity, compared to PLA at the low shear rates. This is associated with the loose percolation structure of GNPs visible in Figure 2c. Above a critical range of shear rates, ($\dot{\gamma }$) ≥ 10 s^−1^ for the 1.5% GNP/PLA and ($\dot{\gamma }$) > 5 s^−1^ for the 9% GNP/PLA, a solid-to-liquid transition associated with shear thinning appears. A very specific behavior for the GNP/PLA systems is that the shear thinning transition is shifted to the lower shear rates if the GNP contents increase and the viscosity slope is much steeper than that of the neat PLA with high negative slope values. As a result, the viscosity of the GNP/PLA nanocomposites in the shear-thinning range becomes a few decades lower than that of the neat PLA, and this effect is more pronounced by increasing the GNP content. Such a viscosity collapse, associated with lubrication, can have notable applications in 3D printing. From a microstructure aspect, the extreme shear thinning of GNP/PLA composites may be driven by rearrangement and alignment of the 2D nanoplatelets parallel to the flow direction, above a shear rate limit [29].

The rheological results of GNP/PLA systems in Figure 3b, combined with the micrographs in Figure 2a,c, confirm the assumption that rigid and nanometrically thin platelets dispersed in the polymer melt can attain a stable alignment in the flow direction for sufficiently strong flows. As predicted by Kamal et al. [33], this orientation effect is due to hydrodynamic slip at the liquid–solid interface. A stable orientation occurs when the hydrodynamic slip length is larger than the thickness of the 2D platelet. The lubrication mechanism can also be attributed to the sliding of GNP nanoplatelets over each other [34,35]. Since the disentanglements of polymer chains and the slip and sliding effects of GNPs are not homogeneous in the melt, this can produce shear banding flow instability resulting in a negative slope of viscosity or shear stress in the shear thinning range.

### 3.3. Rheological Modelling on the Basis of Plate–Plate Experiments

During the plate–plate experiments of the considered nanocomposite melts, the steady shear stress is measured as a function of the shear rate, $\tau =f\left(\dot{\gamma }\right)$, and as a consequence, the viscosity is obtained, $\eta =\tau /\dot{\gamma }$. Usually, the viscosity is modeled by some non-Newtonian models, such as the Power law, the Carreau model and its generalization—the Carreau–Yasuda model, the Casson model, the Cross model, etc. [5,19,36,37,38,39,40,41,42,43,44]. In our last paper [5] the viscosity $\eta (\dot{\gamma })$ of the nanocomposites is fitted with the Carreau–Yasuda viscosity model:(1)$$\eta \left(\dot{\gamma }\right)=\eta_{\infty }+\left(\eta_{0}-\eta_{\infty }\right){\left(1+\lambda^{a}{\dot{\gamma }}^{a}\right)}^{\left(n_{c}-1\right)/a}$$
where $\eta_{0}$ is the zero-shear rate viscosity and $\eta_{\infty }$ is the infinite-shear rate viscosity, λ is the relaxation time and $n_{c}\;$ is the behavior index. However, the values of $n_{c}$ were found to be negative for all the considered materials. This made the further application of (1) for different flows of these materials to be quite complicated for modeling. For example, the analytical Weissenberg–Rabinowitsch–Mooney theory [45] does not apply to the considered here pipe flows, which can be analyzed only by numerical modeling.

In the present paper, we propose an analytical method based on the structure of the registered shear stress during the plate–plate experiment. It occurs that after some value of the shear rate, the stress has a steep slope, as shown in Figure 4 for all discussed materials. Yerushalmi et al. [46] reported that an initially homogeneous shear flow is known to be linearly unstable in any regime, where the fluid’s underlying constitutive curve has a negative slope, $\frac{d\tau }{d\dot{\gamma }}<0$ (See also [27]). Shear banding is regarded as extreme shear thinning with a negative stress slope at high shear rates in the yield stress fluids [25,26]. Therefore, we suspect that the extreme shear thinning of the flow curves in Figure 4 can be attributed to an instability expressed as banding. This phenomenon is well described in [20] and given schematically in Figure 5.

The experiments were performed until the shear rate $\dot{\gamma }\leq 100\;s^{-1}$, as for larger shear rates in the plate–plate rheometer, the flow starts to oscillate. For all discussed materials ${\dot{\gamma }}_{1}<100\;s^{-1}$, but ${\dot{\gamma }}_{2}>100\;s^{-1}$ and is not registered by the experiment. Therefore, we extrapolate the shear stress for higher shear rates than 100 $s^{-1}$ similar to the idea presented in the sketch in Figure 5. An illustration of the banding model for 9 wt.% GNP/PLA is shown in Figure 6, where both shear rates ${\dot{\gamma }}_{1}$ and ${\dot{\gamma }}_{2}$ are denoted, as well as the banding shear stress $\tau_{band}$. In the first stable branch $\dot{\gamma }<{\dot{\gamma }}_{1}$, the shear stress is described by the Herschel–Bulkley model (1), [47]:(2)$$\{\begin{array}{c} \dot{\gamma }=0,\;\;\;\;\;\;\;\;\;\;\;\;\;\;\;\;\;\tau \leq \tau_{0}\\ \tau =\tau_{0}+m{\dot{\gamma }}^{n},\;\;\;\;\;\;\;\;\;\;\;\tau >\tau_{0}\;, \end{array}$$
where $\tau_{0}$ is the yield stress, m and n are constants of the fluid material with dimensions [m] = Pa.$s^{n}$ and n = [-]. 

Grand et al. [48] predicted that the fluid flow is absolutely unstable between the two shear rates in the banding zone, corresponding to the maximum and the minimum shear stress values of the flow curve, while, at other shear rates above the minimum shear stress, the flow is predicted either metastable or stable. In Figure 6, the unstable flow zone of shear rates for an example system, 9 wt.% GNP/PLA, is marked with a box, and it is associated with the start and the end of the shear thinning range.

The values of the model constants $\tau_{0}$, m, n, $\tau_{band}$, ${\dot{\gamma }}_{1}$ and ${\dot{\gamma }}_{2}$ for the considered materials are given in Table 2.

### 3.4. Application of the Rheological Model for 3D Printing Nozzle Flow

A simplified model of the flow of molten nanocomposite during the 3D printing process is proposed, where the experimental nozzle is assumed as a straight tube with a radius R = 0.25 mm. For the measured flow rates, the expected flow velocity in the extruder nozzle-end will not have a parabolic profile due to the complicated rheology, as discussed in the previous section. In [5] the full system of Navier–Stokes equations was solved numerically by the software Ansys/Fluent, with a constant flow rate Q (Table 3) at the ambient pressure at the outlet, without any slip on the nozzle wall. The velocity profile for all materials was found to be similar to a plug flow in almost 90% of the flow region and a boundary layer close to the wall [5].

In the present study, we propose an analytical model in the same geometry of the nozzle, as a straight tube, and an experimentally estimated flow rate Q=m/ρt [5], with the corresponding mean velocity U (Table 3), but coming from the rheological model for the shear stress with banding. It is supposed that the upper banding limit of the shear rate $\dot{\gamma }={\dot{\gamma }}_{2}$ is obtained on the tube wall with r = R. The maximum Reynolds number, $Re_{max}$, close to the tube wall can be found from the minimum viscosity value, at $\dot{\gamma }={\dot{\gamma }}_{2}$, as shown in Table 3. For all discussed cases Remax≪1, which shows that the inertia will not affect the polymer flow, it can be assumed as a simple shear flow with yield (Poiseuille flow) in a tube. However, the flow rates Q, corresponding to such flow in the tube are much smaller than the experimentally measured ones given in Table 3. Therefore, we expect the flow to have a more complicated character and propose the flow region to be split into three regions: a yield stress region, $r\leq R_{0};$ a simple shear region of the tube at $r\leq R_{1}$; and a boundary region close to the wall with constant banding stress $\tau_{band}$, where $R_{1}$ will be obtained from the matching with the flow rate. The following equations are given for the regions:(3)$$\{\begin{array}{c} \dot{\gamma }=0,\;\;\;\;\;\;\;\;\;\;\;\;\;\;\;\;\;\tau_{0}\;\;\;\;\;\;\;\;\;\;\mathrm{at}\;\;\;\;\;\;\;\left(r\leq R_{0}\right)\;\;\\ \frac{1}{r}\frac{d}{\mathrm{dr}}\left(r\tau \right)=P,\;\;\;\;\;\tau_{0}<\tau \leq \tau_{\mathrm{band}}\;\;\;\;\mathrm{at}\;\;\;\left(R_{0}<r\leq R_{1}\right)\;\\ \dot{\gamma }=ar+b,\;\;\;\;\;\;\;\;\tau =\tau_{\mathrm{band}}\;\;\;\;\;\;\;\mathrm{at}\;\;\;\left(R_{1}<r\leq R\right),\; \end{array}$$
where P is the pressure gradient, $R_{0}\;$is the radius of the yield stress region, a and b are the constants:(4)$$P=\frac{2\tau_{\mathrm{band}}}{R_{1}}\;,\;R_{0}=\frac{\tau_{0}R_{1}}{\tau_{\mathrm{band}}}\;,\;a=\frac{{\dot{\gamma }}_{2}-{\dot{\gamma }}_{1}}{R-R_{1}}\;,\;b={\dot{\gamma }}_{2}-\mathrm{aR}.$$

The velocities corresponding to (3) in the different regions are obtained to be:(5)$$\{\begin{array}{c} u_{\mathrm{yield}}=u\left(r=R_{0}\right)\\ u=\frac{n}{n+1}{\left(\frac{P}{2m}\right)}^{\frac{1}{n}}[{\left(R_{1}-R_{0}\right)}^{\frac{1}{n}}-{\left(r-R_{0}\right)}^{\frac{1}{n}}],\\ u_{\mathrm{band}}=\frac{a}{2}\left(r^{2}-R^{2}\right)+b\left(r-R\right) \end{array}$$
and the flow rate becomes:(6)$$W\left(R_{1}\right)={\;\pi R}_{0}^{2}u_{\mathrm{yield}\;}+2\pi \int_{R_{0}}^{R_{1}}u\;\mathrm{rdr}+2\pi \int_{R_{1}}^{R}u_{\mathrm{band}}\;\mathrm{rdr}+{\pi R}_{1}^{2}u_{\mathrm{band}}\left(R_{1}\right)$$

The unknown radius $R_{1}$ is obtained when equating the upper expression to the given flow rate Q:(7)$$W\left(R_{1}\right)=Q.$$

The profiles of the velocity functions (5) in the tube for the different polymers are presented in Figure 7a–e. These profiles have similar forms as those obtained from the numerical model given in [5], although built on the assumption of a complex structure of the flow domain: a simple shear region with yield and a banding region. Since the pressure is not constant, $P_{band}=\frac{2\tau_{band}}{r}$ in the banding region, vortices may exist in it, giving rise to turbulent structures, as discussed in [5]. The materials with yield stress have a pronounced plug flow, Figure 7c–e. The material 9 wt.% GNP/PLA has the smallest $\tau_{band}$ and the minimum$\;{\dot{\gamma }}_{1}$, as seen in Figure 7c and Table 2, thus, the plug velocity region is longer. The drag force coefficient $c_{d}$ of 9 wt.% GNP/PLA, calculated from the shear band $\tau_{band}$ and given in Table 3, has the value 0.67 × 10^4^, which is much smaller than the drag force coefficients of the rest of the materials. Then it can be concluded that 9 wt.% GNP/PLA polymers can be easier printed, which is confirmed also by the experiments of 3D printing. For PLA from Figure 7a, the parabolic velocity profile spreads on a larger zone in the tube, which resembles in some sense the Poseuille profile of Newtonian fluids. This is connected with its ${\dot{\gamma }}_{1}$, which has the maximum value concerning the other materials (Table 2). In our opinion, although the high yield stress zone (Figure 7e), the most difficult material for printing is 9 wt.% MWCNT/PLA. This is due to the highest $\tau_{band}$ (Table 2) and $c_{d}$ (Table 3), which leads to a very high pressure gradient to be achieved in the 3D printer.

## 4. Discussion

The importance of the rheological behavior of polymeric materials in the additive manufacturing materials extrusion process (3D printing ME) is significant for solving problems associated with flow instabilities. In Table 4, a comparison of related works on shear flow rheology, yield stress and shear banding instabilities observed in polymeric materials or complex fluids with graphene-based fillers is presented. In contrast to the related works, the present study investigates the flow instabilities in polymer nanocomposites filled with graphene nanoplatelets and/or carbon nanotubes focused on the 3D printing ME process.

### 4.1. Engineering Design of Printable Hybrid Polymer Nanocomposites

The experimental and analytical flow parameters obtained in this study can be used for designing the rheological complexity of hybrid polymer nanocomposites towards good printability and multifunctionality. Rheological properties, yield stress and shear thinning, are strongly dependent on the shape, size, concentration and dispersion of nanofiller, GNP and MWCNT, in the PLA nanocomposites.

Nanocomposites of PLA filled with two types of GNPs and MWCNTs with different sizes and aspect ratios, as supplied by different providers, are reported in our previous study [7]. It was found that the filler particles with small length and lower aspect ratio (AR) in PLA composite, demonstrate much higher viscosity and worsen printability, compared to the longer particles with a high aspect ratio. Importantly, the properties enhancement by nanofillers is stronger for shorter particles with lower AR. For example, MWCNTs with L = 1.5 μm and AR = 150 in the PLA composite show a decade higher viscosity and worsen printability at 9 wt% filler content, but demonstrate higher elastic modulus and a decade higher enhancement of electrical and thermal conductivity, compared to MWCNTs with L = 10–30 μm and AR = 1000. These essential nanostructure parameters also determine the properties and functionality of the 3D printed parts. Herewith, we discuss how the flow parameters can be used for designing 3D printable hybrid polymer nanocomposites having added functionality.

#### 4.1.1. Yield Stress

The yield stress behavior of PLA nanocomposites incorporating MWCNTs and GNPs can be desirable in 3D printing due to the high extensibility and elasticity below yield that can be important for holding the shape of the 3D-printed parts. The yield stress behavior of the hybrid polymer nanocomposites is associated with the filler–polymer nanostructure above the percolation threshold, which usually ensures added functionality [5,7]. However, the value of the yield stress is a determinant for the processing therefore a limit has to be estimated to ensure a good printability of the nanocomposite material in the additive extrusion. We found in this study, that the nanocomposites with low and moderate yield stress (below 10^3^ Pa), e.g., 9 wt.% GNP/PLA and 1.5 wt.% MWCNT/PLA (Table 2) demonstrate good printability, similar to the neat PLA. In contrast, it is difficult to print samples using 9 wt.% MWCNT/PLA filament, at the same printing conditions, due to the high yield stress (above 10^5^ Pa), which requires a very high pressure to flow in the 3D printing nozzle.

As reported in our previous study [7], at filler contents above the percolation threshold the printable 9 wt.% GNP/PLA and 1.5 wt.% MWCNT/PLA demonstrate multifunctionality. Hybrid nanocomposites have high electrical and thermal conductivity, and electromagnetic shielding efficiency appears, depending on the filler type and concentration. In Figure 8, examples of 3D-printed test samples with a complex shape and good resolution, fabricated using the hybrid nanocomposite filaments of 1.5 wt.% GNP/PLA (a), 9 wt.% GNP/PLA (b) and 1.5 wt.% MWCNT/PLA (c) are presented.

#### 4.1.2. Shear Banding Related to Microstructure

The shear banding flow of PLA nanocomposites with carbonaceous fillers, produced by the alignment, slip and sliding of GNPs in the flow field, as well as the disentanglement of entangled MWCNTs, can have notable applications in 3D printing. Two analytical parameters, banding shear stress,$\;\tau_{band}\;$and drag force coefficient, $c_{d}\;$are proposed here for designing printable hybrid polymer nanocomposites. In the case of GNP/PLA and MWCNT/PLA nanocomposites with 1.5 wt.% filler content, the values of $\tau_{band}\;$(Table 2) and the $c_{d}$ values (Table 3) are slightly higher than that of the neat PLA, which ensure a good printability of the nanocomposites, similar to that of the neat PLA. From a microstructure aspect, the nanofiller arrangement in the 1.5 wt% nanocomposites is around the percolation threshold (of 1.5 wt.% and 3 wt.% for MWCNT and GNP, respectively), which explained the moderate shear stress values [5,7].

However, if consider the 9 wt.% MWCNT/PLA nanocomposite, the dense network structure of entangled MWCNTs, highly above the percolation threshold (Figure 2d), produces a decade higher banding shear stress ($\tau_{band}\;$≥ 10^5^ Pa), and half a decade higher $c_{d}$ than that of PLA, which suppress the flow and worsen the printability. In contrast, the alignment, slip and sliding of GNPs in the flow field in the 9 wt.% GNP/PLA nanocomposite is leading to a collapse of the shear stress with almost four times lower values of $\tau_{band}$ and $c_{d}$ than that of the PLA (Table 2 and Table 3). This strong lubrication effect of GNPs facilitates the printability of nanocomposites, particularly at high filler contents.

Therefore, the control on micro and nanostructure by the type, shape and concentration of the nanofiller can be successfully used to tune the banding shear stress, $\tau_{band}$, and the drag force coefficient, $c_{d}$, towards engineering design of printability and functionality of the hybrid polymer nanocomposites above the percolation threshold.

## 5. Conclusions

In the present study, the shear flow behavior related to the microstructure of hybrid PLA nanocomposite filled with GNP and MWCNT is studied towards application in material extrusion additive manufacturing, 3D printing (ME). Since the disentanglement of polymer chains and the orientation of anisotropic nanofiller in the flow direction are not homogeneous at high shear rates, a highly aligned phase next to a less aligned one coexists in the nanocomposite melt flow, that is regarded to undergo flow instability. As a result, the measured steady-state shear stress of the filled nanocomposites GNP/PLA and MWCNT/PLA as a function of the shear rate has a complicated behavior compared to the pure polymer. These hybrid polymer nanocomposites can be classified as extremely shear-thinning and modeled as yield stress fluids with banding. The flow in the nozzle is studied by a simple analytical model considering the flow region composed of three different regions in the tube, which match on their boundaries. This model has an advantage over the complex numerical model used in our previous study, as it explains the structure of the flow in the nozzle tube and makes it possible to conclude which materials are easier or harder to be printed. Three flow parameters, the yield stress, banding shear stress and drag force coefficient are determined from the experimental rheological results and analytical model, and proposed for the engineering design of hybrid polymer nanocomposites with good printability and added functionality.

## Figures and Tables

**Figure 1 nanomaterials-13-00835-f001:**
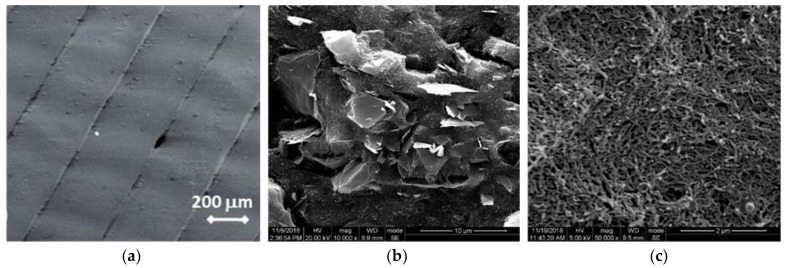
SEM micrographs of: (**a**) the surface of 3D printed GNP/PLA, (**b**) cross-section of GNP/PLA, and (**c**) cross-section of MWCNT/PLA at 9 wt.% filler content.

**Figure 2 nanomaterials-13-00835-f002:**
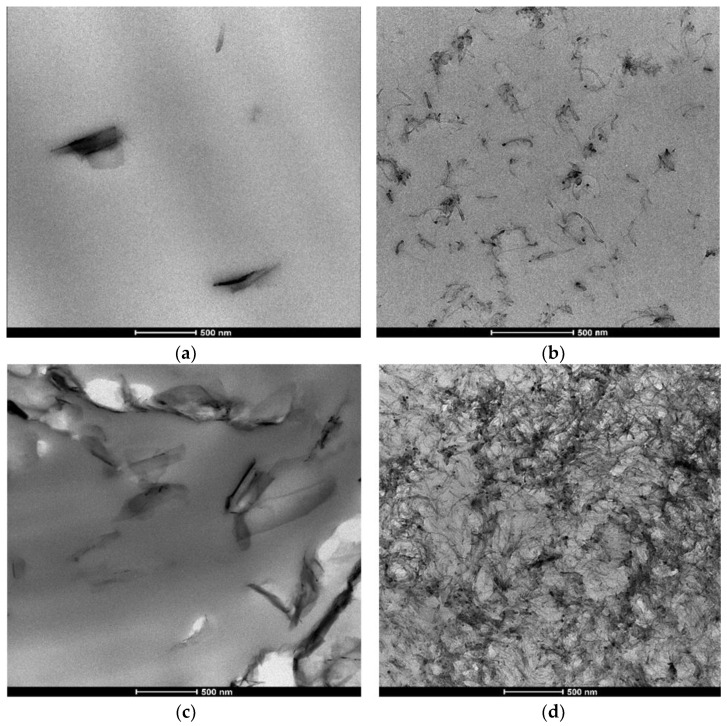
TEM micrographs of the hybrid nanocomposites: (**a**) 1.5 wt.% GNP/PLA, (**b**) 1.5 wt.% MWCNT/PLA; (**c**) 9 wt.% GNP/PLA, and (**d**) 9 wt.% MWCNT/PLA, at the same magnification.

**Figure 3 nanomaterials-13-00835-f003:**
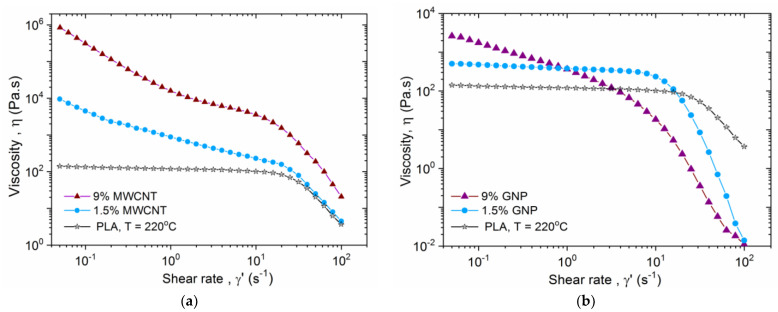
Viscosity vs. shear rate of (**a**) MWCNT/PLA and (**b**) GNP/PLA nanocomposites at filler contents of 0, 1.5 and 9 wt.%, at 220 °C.

**Figure 4 nanomaterials-13-00835-f004:**
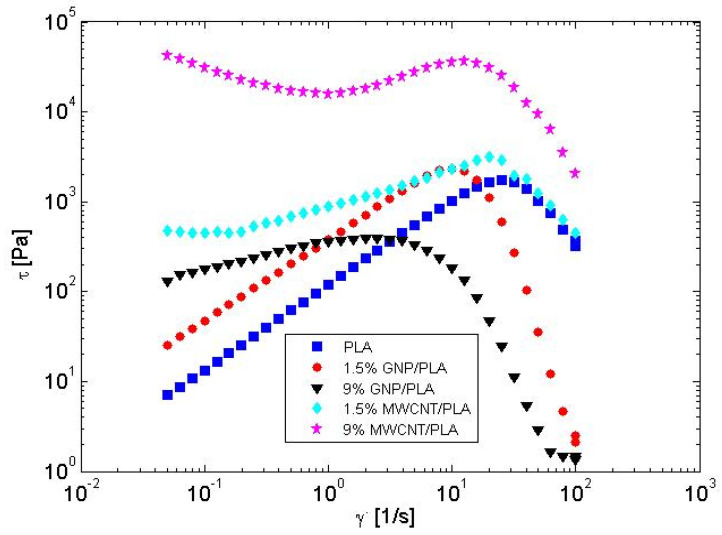
Experimental data for the shear stress as a function of the shear rate.

**Figure 5 nanomaterials-13-00835-f005:**
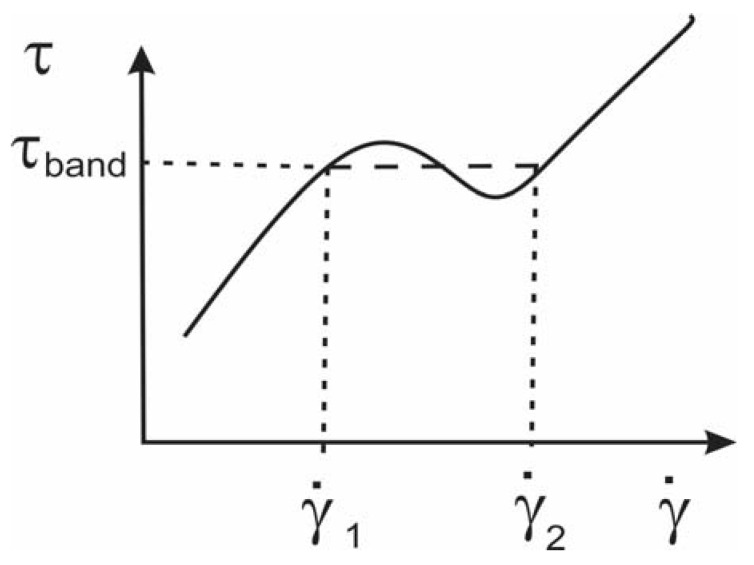
Schematic flow curve, showing the two stable branches with $\dot{\gamma }<{\dot{\gamma }}_{1}$ and $\dot{\gamma }>{\dot{\gamma }}_{2}$. In the unstable region ${\dot{\gamma }}_{2}>\dot{\gamma }>{\dot{\gamma }}_{1}$ the stress is assumed constant $\tau =\tau_{band}$.

**Figure 6 nanomaterials-13-00835-f006:**
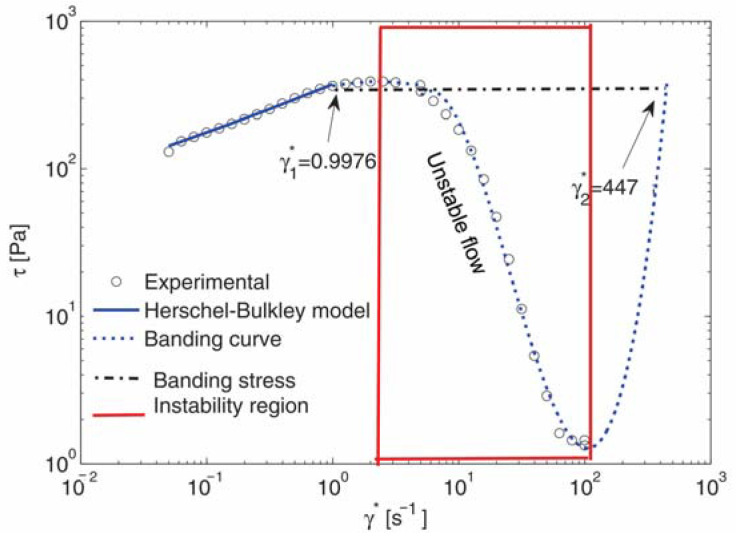
The shear stress modeling of 9 wt.% GNP/PLA: experimental data (o); banding curve with a blue dotted line; Herschel–Bulkley model with a solid blue line; shear stress as the constant $\tau_{band}=\tau \left({\dot{\gamma }}_{1}\right)=\tau \left({\dot{\gamma }}_{2}\right)$ with a dash-dotted black line. The unstable flow region between the maximum and minimum shear stress and the corresponding shear rates is marked as a red box, i.e., unstable flow with negative shear stress gradient.

**Figure 7 nanomaterials-13-00835-f007:**
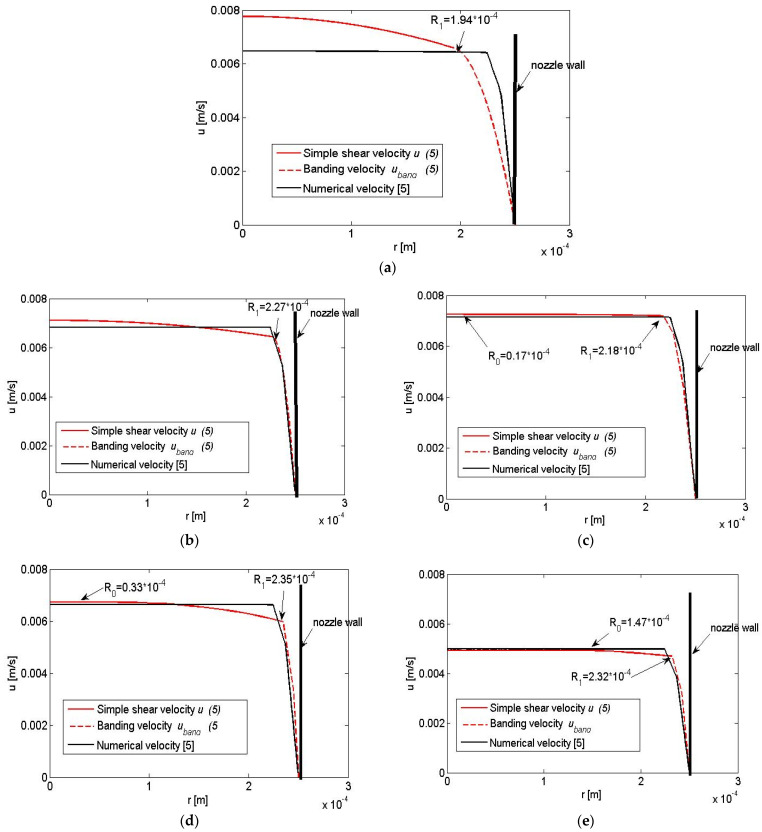
Comparison between the velocities’ profiles in the yield and simple shear regions (solid red line) and in the banding region (dashed red line) with the numerical velocity [5], for (**a**) PLA. (**b**) 1.5% GNP/PLA and (**c**) 9% GNP/PLA. (**d**) 1.5% MWCNT/PLA and (**e**) 9% MWCNT/PLA.

**Figure 8 nanomaterials-13-00835-f008:**
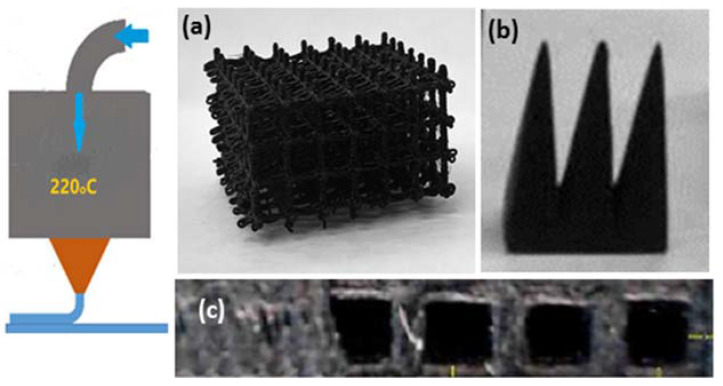
3D printed complex structures, at 220 °C, nozzle D = 0.5 mm using hybrid nanocomposite filaments: (**a**) 1.5 wt.% GNP/PLA; (**b**) 1.5 wt.% MWCNT/PLA; and (**c**) 9 wt.% GNP/PLA.

**Table 1 nanomaterials-13-00835-t001:** Characteristics of the nanofillers, GNP and MWCNT.

Filler Type	GNP	MWCNT
Trade name	TNGNP	NC7000
Producer	Times Nano, China	Nanocyl, Belgium
Carbon purity, wt%	>99.5	90
Layers, number	<20	-
Thickness, nm	4–20	-
Mean diameter, μm	5–10	-
Outer diameter, nm	-	9.5
Length, μm	-	1.5
Aspect ratio	500	150
Surface area (SSA), m^2^/g	30–40	250–300
Functionalization	-	oxidized

**Table 2 nanomaterials-13-00835-t002:** Rheological and banding properties.

Material	$\tau_{0}$ [Pa]	$m\;[\mathrm{Pa}.s^{n}]$	n [-]	${\dot{\gamma }}_{1}\;[s^{-1}]$	${\dot{\gamma }}_{2}\;[s^{-1}]$	$\tau_{\mathrm{band}}$ [Pa]
PLA	0	118.85	0.9441	12.56	223	1295.85
1.5% GNP/PLA	0	380.19	0.9021	6.295	552	1998.82
9% GNP/PLA	29.71	343.1	0.37	0.9976	447	372.5
1.5% MWCNT/PLA	334.7	511.6	0.6054	9.976	772	2393.9
9% MWCNT/PLA	15,880	488.4	2.124	3.972	518	25,022.63

**Table 3 nanomaterials-13-00835-t003:** Flow and material parameters in the nozzle.

Material	Q [mm^3^/s]	U [mm/s]	ρ [kg/m^3^]	$\mu_{2}=\frac{\tau_{\mathrm{band}}}{{\dot{\gamma }}_{2}}[\mathrm{Pa}.s]$	${Re}_{\max}=\frac{\rho UR}{\mu_{2}}$	$c_{d}=\frac{\tau_{\mathrm{band}}}{\rho U^{2}}$
PLA	1.1723	5.97	1240	5.81	3.2 × 10^−4^	2.9 × 10^4^
1.5% GNP/PLA	1.2417	6.32	1250	3.62	5.4 × 10^−4^	4 × 10^4^
9% GNP/PLA	1.3004	6.62	1260	0.83	25 × 10^−4^	0.67 × 10^4^
1.5% MWCNT/PLA	1.2064	6.14	1250	3.10	6.2 × 10^−4^	5 × 10^4^
9% MWCNT/PLA	0.9060	4.61	1260	48.3	0.3 × 10^−4^	93 × 10^4^

**Table 4 nanomaterials-13-00835-t004:** Comparison of related works.

Materials	Shear Flow Rheology	Yield	Shear Banding	3D Printing (ME)	References
Polymer solutions	yes	no	yes	no	[20,22,23]
Wormlike micelles	yes	no	yes	no	[24,25]
All types of polymers	yes	yes	yes	no	[26,27]
Suspensions with graphene oxide (GO)	yes	yes	yes	no	[29]
Liquid crystalline suspensions with GO	yes	no	yes	no	[30]
Polymer melts	yes	yes	yes	yes	[2,19,28]
Polymer melts with GNPs and/or MWCNTs	yes	yes	yes	yes	present work

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
