# Peer review of "Exploring Effects of Graphene and Carbon Nanotubes on Rheology and Flow Instability for Designing Printable Polymer Nanocomposites"

_nanomaterials, 2023, doi:10.3390/nano13050835_

Round 1
Reviewer 1 Report
In this paper, the authors discussed the hydrodynamic characteristics of hybrid polylactic acid (PLA) nanocomposites filled with graphene nanoplates (GNP) and multi-walled carbon nanotubes (MWCNT) to prepare multifunctional filaments for 3D printing. The measured shear stress by a plate-plate rheometer of PLA, 1.5% and 9% GNP/PLA and MWCNT/PLA shows an instability for high shear rates, which is expressed as shear banding. The present model gives an insight into the flow structure and better explains the reasons for printing enhancement. Experimental and modelling parameters are explored in designing printable hybrid polymer nanocomposites with added functionality. I believe that publication of the manuscript may be considered only after the following issues have been resolved.
1. In order to better highlight the advantages of this work, the author needs to provide a table to compare related work.
2. The scale in Fig. 2b is not clear and needs to be adjusted.
3. The introduction of graphene and evasive carbon nanotubes can improve the performance of the composite system. What is the physical mechanism? The author needs to discuss and analyze in the abstract and the text.
4. The introduction can be improved. The articles related to some applications of graphene materials should be added such as Sensors 2022, 22, 6483; ACS Sustain. Chem. Eng. 2015, 3, 1677–1685; Diamond & Related Materials 128 (2022) 109273; Talanta 2015, 134, 435–442.
5. Please check the grammar and spelling mistakes of the whole manuscript.
Reviewer 2 Report
In this paper, the authors study the rheological properties related to the microstructure of hybrid poly(lactic) acid (PLA) nanocomposites, filled with graphene nanoplatelets (GNP) and multiwall carbon nanotubes (MWCNT), to produce multifunctional filament for 3D printing.
The subject is timely and the presented results are interesting for practical applications. The proposed model is reasonable.
The manuscript could be ready for publication after a revision:
“Hybrid polymer nanocomposites incorporating 1D and 2D carbonaceous nanofillers are intensively studied for the development of multifunctional materials for 3D printing application with robust electrical, electromagnetic, thermal and mechanical properties [7, 8].” I suggest adding to 7-8 the following paper https://doi.org/10.1016/j.compositesb.2017.10.020 dealing with a polymer filled by different 1D and 2D carbonaceous nanocomposites.
It seems that three classes of samples were prepared, with 0, 1.5, and 9 wt.% filler content. The authors could clarify the reason for this choice. Are 1.5 and 9 wt.% extreme cases of the filler content?
“The 1.5 % CNP/PLA micrograph (a) 155 shows partly exfoliated graphene particles, not contacting each other, which is a typical structure before the percolation threshold. While the microstructure of 1.5% MWCNT/PLA (b) consists of well-dispersed single nanotubes and small aggregates which 158 are contacting to form a percolation network.” How can the authors be sure that the samples are below or above the percolation threshold? I wonder if they performed any electrical measurements to confirm it.
“Rheological properties, yield stress and shear thinning, are strongly dependent on the shape, size, concentration and dispersion of nanofiller, GNP and MWCNT, in the PLA nanocomposites.” I fully agree with this statement. The authors have used GNP and MWCNT of the given size, as specified in table 1. It would be interesting to compare samples obtained with GNP or MWCNT of different sizes, perhaps from a different provider. Which would be the effect of longer MWCNTs?
Round 2
Reviewer 1 Report
Accept in present form.